# Complications of Central Venous Access Devices Used in Palliative Care Settings for Terminally Ill Cancer Patients: A Systematic Review and Meta-Analysis

**DOI:** 10.3390/cancers15194712

**Published:** 2023-09-25

**Authors:** Clement Chun-Him Wong, Horace Cheuk-Wai Choi, Victor Ho-Fun Lee

**Affiliations:** 1LKS Faculty of Medicine, The University of Hong Kong, Hong Kong, China; ccw19@connect.hku.hk; 2School of Public Health, LKS Faculty of Medicine, The University of Hong Kong, Hong Kong, China; 3Department of Clinical Oncology, Centre of Cancer Medicine, School of Clinical Medicine, LKS Faculty of Medicine, The University of Hong Kong, Hong Kong, China

**Keywords:** central venous access devices, central line, peripherally inserted central catheter, totally implanted port, complications, systematic review, meta-analysis

## Abstract

**Simple Summary:**

Common central venous access devices (CVADs) include peripherally inserted central catheters (PICCs), central lines, and totally implanted ports (PORTs). Studies have shown PORTs have the least catheter-related complications, while PICCs are associated with the highest risk. The delivery of systemic anticancer therapy is known to be a risk factor in CVAD-related complications. Little is known about the complication profile of each CVAD when it is used for purely palliative intent without systemic anticancer therapy. We performed a systematic review and meta-analysis evaluate the complication rates of different types of CVADs used in palliative care settings. The use of central lines in terminally ill cancer patients was found to have fewer complications than PICCs in terms of overall complications, catheter-related bloodstream infection, and thromboembolism. No conclusion can be made on PORTs due to a lack of publications. As there are currently no clinical guidelines regarding the choice of CVAD in terminally ill cancer patients, our study summarizes the current available evidence to provide a basis for further studies on this important issue.

**Abstract:**

(1) Background: Central venous access devices (CVADs) have been commonly employed during various courses of anticancer treatment. Currently, there are a few types of clinically available CVADs, which are associated with short-term and long-term complications. However, little is known about the complication rates when CVADs are used only in palliative care settings. We therefore performed a systematic review and meta-analysis of all the published literature to evaluate the complication rates of CVADs in this clinical setting. (2) Methods: A systematic review and meta-analysis were conducted to identify publications from PubMed/MEDLINE, Embase (Ovid), Scopus, Cochrane Library, CINAHL, Google Scholar, and trial registries. Publications reporting the complication rates of PICCs, central lines, and PORTs in palliative settings for terminally ill cancer patients were included, while those on the use of systemic anticancer therapy and peripheral venous catheters were excluded. The outcome measures included overall complication rate, rate of catheter-related bloodstream infection (CRBSI), and rate of thromboembolism (TE). This systematic review was registered with PROSPERO (CRD42023404489). (3) Results: Five publications with 327 patients were analyzed, including four studies on PICCs and one study on central lines. No studies on PORTs were eligible for analysis. The overall complication rate for PICCs (pooled estimate 7.02%, 95% CI 0.27–19.10) was higher than that for central lines (1.44%, 95% CI 0.30–4.14, *p* = 0.002). The risk of CRBSI with PICCs (2.03%, 95% CI 0.00–9.62) was also higher than that with central lines (0.96%, 95% CI 0.12–3.41, *p* = 0.046). PICCs also had a trend of a higher risk of TE (2.10%, 95% CI 0.00–12.22) compared to central lines (0.48%, 95% CI 0.01–2.64, *p* = 0.061). (4) Conclusions: PICCs for palliative cancer care were found to have greater complications than central lines. This might aid in the formulation of future recommendation guidelines on the choice of CVAD in this setting.

## 1. Introduction

Central venous access devices (CVADs) are used pervasively in cancer patients across different stages of anticancer treatment. They are needed in early stages for the administration of chemotherapy to reduce the risk of extravasation and local toxicity [1]. In advanced stages, they are inserted for palliative care, including the transfusion of blood products, intravenous fluid replacement, and total parenteral nutrition (TPN) [2]. Commonly applied CVADs include peripherally inserted central catheters (PICCs), central lines, and totally implanted ports (PORTs) [2,3]. Among these options, the insertion of a PICC can be performed as a bedside procedure, and its low cost makes it easily accessible [3,4]. In contrast, a PORT is the most technical demanding and expensive option, leading to its low usage [5].

Previous studies primarily focused on the delivery of systemic anticancer therapy with the three listed devices in terms of efficacy and complications. Complications can be further divided into early complications during insertion and late complications after the first delivery through the catheters. The most frequent early complication is pneumothorax, and others include air embolism, arterial puncture, and arrhythmia [6]. Late complications have been the major parameters for comparisons in the literature, and they consist of catheter-related bloodstream infection (CRBSI), thromboembolism, and mechanical failure [6,7]. A randomized controlled trial suggested that PORTs were superior to PICCs and central lines, in which the complication rate was reduced by 14–15% to 29–32% [8]. However, the non-inferiority of PICCs to central lines was not confirmed, possibly due to inadequate power. In another two meta-analyses, the superiority of PORTs over PICCs was substantiated, demonstrating a reduction in thromboembolism by 4.2% and a reduction in malfunctions by 3.6% [9,10].

Nevertheless, it was identified that systemic anticancer treatment is a well-recognized risk factor for CVAD-related thrombosis [11,12,13]. In particular, cytotoxic or immunosuppressive chemotherapy elevated the risk of cancer-related venous thromboembolism by 50% when compared to cancer patients with no treatment prescribed [14]. Therefore, the complication profiles can be totally different when the CVADs are inserted purely for palliative intent without the administration of chemotherapy. Unfortunately, there is very limited evidence of the use of CVADs in purely palliative care settings. The clinical decision of which type of CVAD is to be inserted for palliation varies across centers due to the lack of evidence-based guidelines. In view of the above, this systematic review was conducted to compare the CVAD-related complications among the three different types of CVADs, namely PICCs, central lines, and PORTs, for palliative care in terminally ill cancer patients.

## 2. Materials and Methods

### 2.1. Search Strategy and Selection Criteria

We performed a systematic review and literature search for articles published from inception to 30 April 2023 using PubMed-MEDLINE, Embase (Ovid), Scopus, Cochrane Library, CINAHL, Google Scholar, as well as trial registries, in accordance with the Preferred Reporting Items for Systematic Review and Meta-analysis (PRISMA) guidelines [15]. The search terms included “central venous catheter OR central venous access OR central line OR Hickman OR Hickman catheter OR Port-a-cath OR PORT catheter OR PICC OR peripherally inserted central catheter”, and “cancer OR cancers”, and “palliative care OR terminally ill OR critically ill”. The detailed search strategy and the PICOS framework are described in the Appendix A. We registered our study with the International Prospective Register of Systematic Reviews (PROSPERO) (CRD42023404489).

Our search focused on all full-length articles, case reports, and case series published in the English literature. Palliative care as the most important inclusion criterion of our systematic review refers to best supportive care, conservative management with or without the use of total or peripheral parenteral nutrition, intravenous fluid replacement, blood product transfusion, and the administration of intravenous drugs other than systemic anticancer therapy. Publications describing the use of CVADs for administering any forms of systemic anticancer therapy (including but not limited to chemotherapy, targeted therapy, hormonal therapy, immune checkpoint inhibitor, cellular therapy, intravenous or endovascular radioisotope or radioligand therapy, etc.) or peripheral venous catheters, and those written in a non-English language were excluded. Abstracts and full-text publications were reviewed by two authors (H.C.-W.C. and V.H.-F.L.) independently, and disagreement was resolved by consensus. In the case of having unknown or missing information in the published studies, emails were sent to the corresponding authors for clarification. Ethics approval was waived as this was a systematic review and meta-analysis without subject recruitment or the use of any identifiable patient data.

### 2.2. Quality Assessment of Selected Studies

We referred to the Newcastle–Ottawa Scale (NOS) to assess the quality of the selected studies (https://www.ohri.ca/programs/clinical_epidemiology/oxford.asp, accessed on 1 May 2023). The NOS assess the quality of the studies based on three domains, namely (a) the selection of the study groups, (b) the comparability of the groups, and (c) the ascertainment of the outcome of interest for cohort studies. In general, a study having a score of 7 or above, out of a maximum of 9, would be considered of high quality with a low risk of bias. Studies with a score of 6 or below may represent a considerable risk of bias.

### 2.3. Statistical Analysis

The data extracted for further analysis of this meta-analysis included patient demographics, cancer type, indication of the CVAD, and complication rates arising from the use of CVADs. Only the complications that were reported in all of the selected papers were included for subsequent statistical analysis. The complication rates used as outcome measures in this meta-analysis included catheter-related bloodstream infection (CRBSI), thromboembolism (TE), and overall complication rates (including both CRBSI and TE), since they are the most commonly reported and clinically important complications associated with CVAD insertion. The pooled estimates of the CVAD-related complications were illustrated by forest plots, and comparisons among different CVADs were performed. A leave-one-out meta-analysis was also performed to examine influential studies on the pooled estimate [16]. Heterogeneity among publications was assessed via I^2^ and τ^2^ tests. The random effects model was used when I^2^ > 50% or *p* value < 0.1 in the τ^2^ test, which suggested potential heterogeneity. Otherwise, the fixed effect model was adopted. Data analysis was performed with statistical software R (version: 4.3.0) with Rstudio (version: 2023.03.1+446). The meta and metafor packages were used for preparing the meta-analyses. Statistical significance was defined as *p* value < 0.05 (two-sided).

## 3. Results

### 3.1. Search Strategy Results

Initially, 41 publications were identified from the literature search and review (Figure 1). Thirty-three articles were excluded during the first stage of screening, including seven review articles or guidelines without descriptions of any patient data, twelve articles for non-palliative intent, four articles with peripheral catheters and ten articles without details of CVAD-related complications. After screening the abstracts, eight publications were chosen to proceed with full-text review and two papers were excluded due to incomplete data retrieval [17,18]. A paper studying PORTs was excluded due to the administration of chemotherapy in a palliative care setting [19]. Hence, a total of five publications were included with four studies on PICCs [20,21,22,23] and one study on central lines [24]. The inter-rater reliability of publication screening by the authors was 97.6%, which was related to the doubtful eligibility of one publication, resolved by consensus.

Based on the Newcastle–Ottawa Scale (NOS), the selected studies were generally awarded quality scores ranging from 7 to 9, suggesting that the selected studies were of high quality (Table 1).

### 3.2. Baseline Patient Characteristics

The characteristics of 327 patients analyzed in this meta-analysis are shown in Table 2. Regarding the type of CVAD, 207 patients who received central line insertion were contributed by a single study, and the remaining 120 patients who had PICCs inserted were from the remaining four publications [20,21,22,23,24]. There was a total of 209 central line insertions in 207 patients due to 2 failed insertions [24]. The most common primary cancer was gastrointestinal cancer (47.7%), followed by genitourinary cancer (14.1%). As a patient can have multiple indications for CVAD insertion, the indications were stated accordingly. Total parenteral nutrition and intravenous fluid replacement were the most frequent indications, with percentages of 84.7% and 75.5%, respectively. The 13 undetermined patients in terms of sex, types of primary cancer, and indication of CVAD were from Bortolussi et al. [22]. The study combined the patient demographics with midline catheters but the number and complication rate of PICCs were reported independently. Hence, it was decided to include this paper, as the primary outcome we were measuring was stated clearly in this study.

### 3.3. Complication Rates between Central Line and PICC

CRBSI and TE were identified to be universally reported in the five studies. The overall complication rate including both CRBSI and TE yield a pooled estimate of 7.02% (95% CI 0.27–19.10) for PICCs, which was higher than 1.44% (95% CI 0.30–4.14, *p* = 0.002) for central lines (Figure 2). PICCs had a higher complication rate of CRBSI (pooled estimate 2.03%, 95% CI 0.00–9.62) when compared to central lines (pooled estimate 0.96%, 95% CI 0.12–3.41%, *p* = 0.046) (Figure 3). PICCs also tended to have a higher risk of TE (pooled estimate 2.10%, 95% CI 0.00–12.22) than central lines (pooled estimate 0.48%, 95% CI 0.01–2.64%, *p* = 0.061) (Figure 4).

In addition to the major complications, some minor complications were reported in two studies on PICCs. There were two cases of thrombophlebitis, eight cases of bleeding, and seven cases of self-removal among 68 patients [20,21].

## 4. Discussion

To the best of our knowledge, this is the first systematic review and meta-analysis to investigate the complications of CVADs in a palliative care setting for terminally ill cancer patients. Previous studies included the delivery of systemic anticancer therapy as an indication for CVADs, which could substantially affect the complication profiles of CVADs in palliative care. With reference to the above findings, it was observed that generally, PICCs had a higher rate of complications in all aspects when compared to central lines. Moreover, central lines were shown to be more cost-effective as the relatively high incidence of complications using PICCs was associated with a higher cost for maintenance [25]. Central lines can thus provide a better safety profile at a lower cost [26].

As regards the complication rates of CVADs in palliative care settings, the values were seemingly lower than those of CVADs delivering chemotherapy or other systemic anticancer therapy. The greatest value was recorded to be 7.02% for the overall complications with PICCs. A prospective observational study of CVADs with chemotherapy delivery yielded an overall complication rate of 30.1%, infection rate of 12.8%, and thrombotic rate of 4.5% [27]. It appeared that the use of chemotherapy is associated with a higher risk of complications, but such a conclusion has yet to be drawn. Hence, further studies are required in order to delineate the detrimental effect of chemotherapy in CVADs. The more frequent complications may also be attributed to the chemotherapy-induced immunocompromised state, rendering patients more prone to infections. Moreover, both cytotoxic chemotherapy and targeted agents mediate or induce damages to the endothelial layer of blood vessels, contributing to a higher chance of thromboembolism [28,29,30].

In view of the limited dataset in the meta-analysis, a leave-one-out meta-analysis was performed for PICCs to test the robustness of the pooled estimates (Appendix A) [16]. A publication is considered to be influential when its removal leads to the pooled estimate to fall outside the range of the 95% confidence interval [16]. The relevant study should be screened to look for an explanation for the discrepancies, which include the study design and subject selection. With reference to the overall complication rates, the pooled estimate when omitting the study by Park et al. was 13.09%, which was outside the 95% CI (2.41 to 12.34%) under the common effect model (Appendix A). Likewise, it was noticed that the pooled estimate of TE without the study by Yamada et al. (0.00%) was outside the 95% CI (0.13 to 6.96%) under the common effect model (Appendix A). This was because the study by Yamada et al. was the only one that reported TE, with seven cases out of 39 insertions [23]. No CRBSI or TE was reported in this study, hence contributing to the above finding [21]. These papers were not identified as outliers as the values did not deviate substantially from the 95% CIs. The overall results were not solely dependent on a single study, and thus, the robustness was preserved.

Despite the consistent findings with the previous studies regarding CVADs, there were limitations in our current study. Firstly, there was no publication reporting PORTs used in palliative settings without chemotherapy administration. Therefore, we were only able to make comparisons between PICCs and central lines. Traditionally, the insertion of a PORT often requires general anesthesia or intravenous sedation, and terminally ill cancer patients are not suitable candidates [31]. However, recent studies reported that the placement of a PORT can be achieved via a superficial cervical plexus block with local anesthesia, which is the same technique used in the insertion of central lines [32,33]. It is expected that PORTs will gain increasing popularity and relevant studies will be published soon. Our analysis was also limited by the small number of publications conducted in developed countries in Asia and Europe, leading to a high level of heterogeneity and a relative lack of generalizability to other regions where CVAD insertion may not be so popular and readily available. In addition, all the data for central line insertion originated from a single study, and no comparison within the same device could be made. Moreover, the sample sizes of the four selected studies which investigated PICCs were fewer than 40. Some studies did not observe CRBSI or TE, and the corresponding (pooled) confidence intervals may have included 0. This small sample size issue could have been a concern, as every observed event may have affected the outcome substantially (e.g., the proportion could have been changed by 5% when the sample size was 20). The findings should be interpreted with caution. Furthermore, the overall complication rate only took CRBSI and TE into account, leaving the other less common complications out of the analysis, since the latter were rarely reported, subject to the investigators’ decision. This might have led to the underestimation of the overall complications. Finally, we were not able to identify if other patient parameters and comorbidities like body weight, diabetes mellitus, and hyperlipidemia, previously reported as associated factors of CVAD-related infections, also played a role in CVAD-related complications in our current meta-analysis, as they were usually considered less clinically relevant and thus rarely reported in the literature [34]. Nonetheless, our study has provided insights for clinicians when choosing the most suitable CVAD for terminally ill cancer patients for palliative care. The expected patient life expectancy, the expertise of the treating physicians, the availability of facilities and resources for CVAD insertion in a palliative clinic or hospice setting, and the cost incurred by CVAD insertion and its cost-effectiveness are also important considerations in the real-world setting, which should also be meticulously deliberated and thoroughly discussed with patients and their caretakers. In order to formulate recommendation guidelines for CVADs in palliative care, further research is warranted to investigate the complication profiles of various types of CVADs.

## 5. Conclusions

The use of central lines in terminally ill cancer patients was found to have fewer complications than the use of PICCs in our study. No conclusion could be made for PORTs because of a paucity of relevant publications. Future studies are warranted to compare the complications related to all the currently available types of CVADs in terminally ill cancer patients.

## Figures and Tables

**Figure 1 cancers-15-04712-f001:**
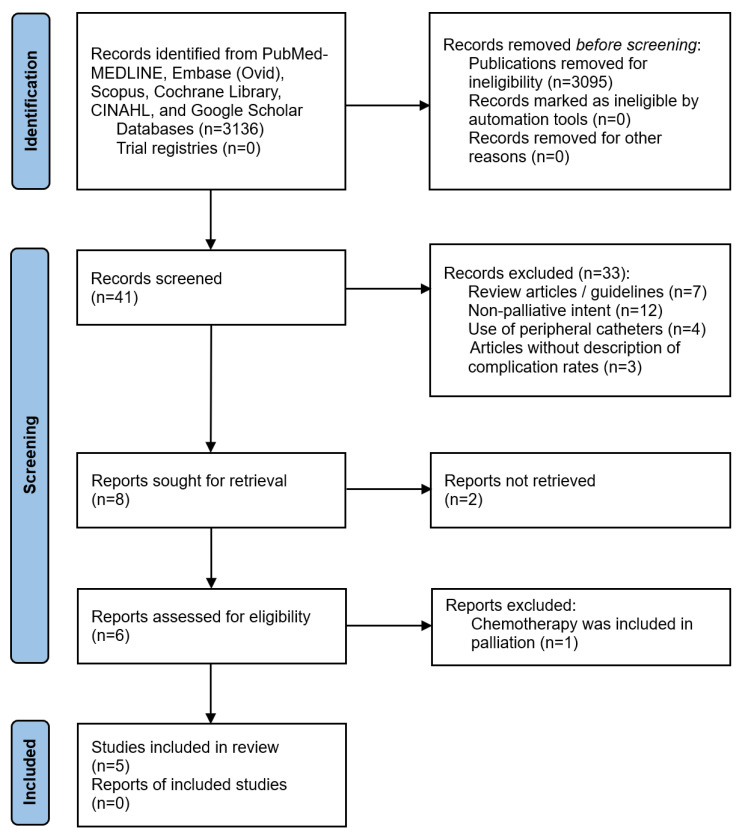
PRISMA flow chart showing the identification and selection process of included studies for analysis.

**Figure 2 cancers-15-04712-f002:**
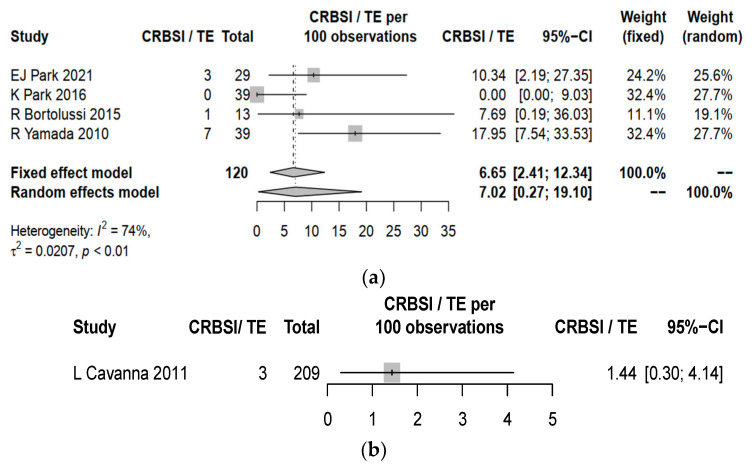
Forest plot for the rates of catheter-related bloodstream infection (CRBSI) and thromboembolism (TE) in (**a**) PICCs and (**b**) central lines. Comparison between pooled estimates: *p* = 0.002, standard error = 3.11. EJ Park 2021 [20], K Park 2026 [21], R Bortolussi 2015 [22], R Yamada 2010 [23] and L Cavanna 2011 [24].

**Figure 3 cancers-15-04712-f003:**
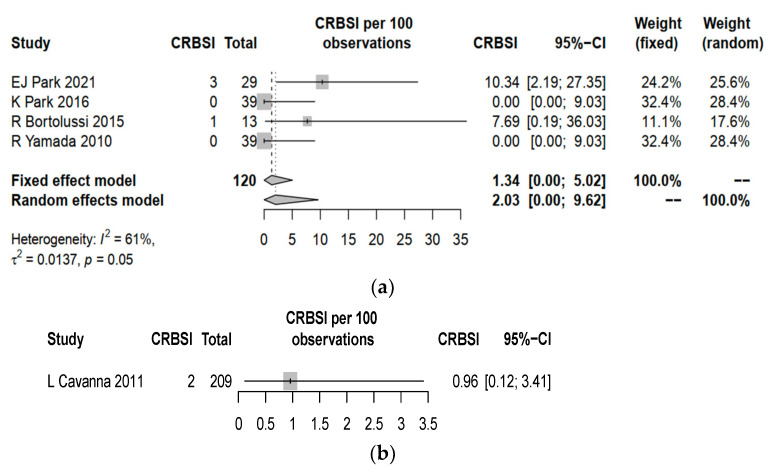
Forest plot for the rates of catheter-related bloodstream infection (CRBSI) in (**a**) PICCs and (**b**) central lines. Comparison between pooled estimates: *p* = 0.046, standard error = 1.99. EJ Park 2021 [20], K Park 2026 [21], R Bortolussi 2015 [22], R Yamada 2010 [23] and L Cavanna 2011 [24].

**Figure 4 cancers-15-04712-f004:**
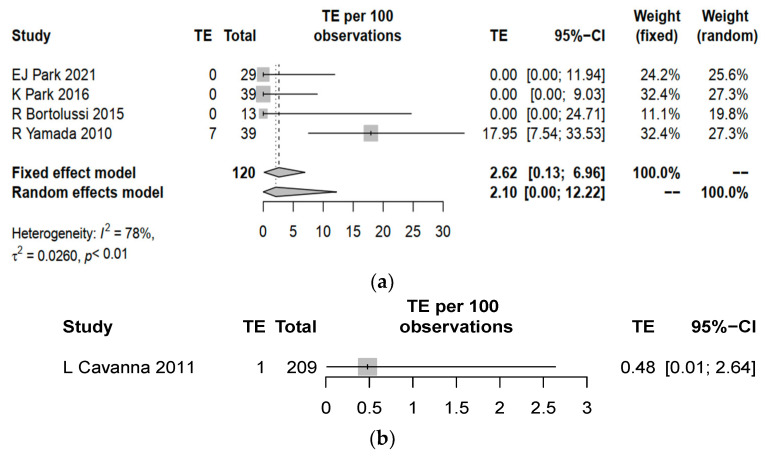
Forest plot for the rates of thromboembolism (TE) in (**a**) PICCs and (**b**) central lines. Comparison between pooled estimates: *p* = 0.061, standard error = 1.87. EJ Park 2021 [20], K Park 2026 [21], R Bortolussi 2015 [22], R Yamada 2010 [23] and L Cavanna 2011 [24].

**Table 1 cancers-15-04712-t001:** Quality assessment of selected studies with Newcastle–Ottawa Scale.

Study	EJ Park, 2021 [20]	K Park, 2016 [21]	R Bortolussi, 2015 [22]	R Yamada, 2010 [23]	L Cavanna, 2011 [24]
Selection					
1. Representativeness of the exposed cohort	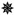	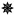	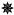	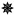	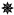
2. Selection of the non-exposed cohort	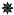		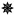		
3. Ascertainment of exposure	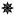	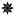	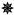	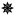	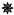
4. Demonstration that outcome of interest was not present at start of study	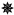	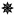	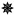	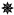	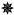
Comparability					
1. Comparability of cohorts on the basis of the design or analysis	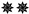	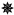	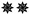	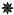	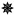
Outcome					
1. Assessment of outcome	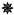	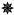	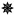	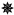	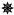
2. Was follow-up long enough for outcomes to occur	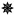	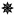	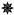	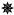	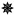
3. Adequacy of follow up of cohorts	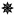	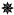	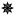	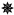	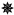
Score	9	7	9	7	7

A maximum of two 
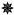
 can be given for comparability.

**Table 2 cancers-15-04712-t002:** Distribution of predominant microscopic morphological features.

Characteristics	Number of Patients (%) Total (*n* = 327)
Mean age (range)	68.4 (22–89)
Sex	
Male	180 (55.0)
Female	134 (41.0)
Undetermined	13 (4.0)
Type of primary cancer	
Gastrointestinal	156 (47.7)
Hepatobiliary and pancreatic	41 (12.5)
Genitourinary	46 (14.1)
Other solid tumors	64 (19.6)
Hematological	7 (2.1)
Undetermined	13 (4.0)
Indication of CVAD	
Total parenteral nutrition	277 (84.7)
Intravenous fluid replacement	247 (75.5)
Intravenous medication	45 (13.8)
Blood product transfusion	12 (3.67)
Undetermined	13 (4.0)
Type of CVAD	
PICC	120 (36.7)
Central line	207 (63.3)

CVAD, central venous access device; PICC, peripherally inserted central catheter.

## Data Availability

All articles in this manuscript are available from MEDLINE, Embase, Cochrane Library, CINAHL, and Google Scholar.

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
