# Peer review of "Complications of Central Venous Access Devices Used in Palliative Care Settings for Terminally Ill Cancer Patients: A Systematic Review and Meta-Analysis"

_cancers, 2023, doi:10.3390/cancers15194712_

Round 1

Reviewer 1 Report

Dear Authors,

I read your work entitled “Complications of Central Venous Access Devices Used in Palliative Care Setting for Terminally-ill Cancer Patients: A Systematic Review and Meta-analysis” and here I enclose my recommendations to you:

1.     The introduction is simple, short and clearly written and very nicely reports the existing knowledge around this topic.

2.     The methodology is asserted since the Authors work was approved by PROSPERO. In order the Authors to enhance their work I suggest them to enter in their methodology the PICOS system as well.

3.     It would be good for the reader if the Authors would briefly describe the detection of the articles as they are in prism because they start the description from the 41 that reviewed and do not give details about the first part like the 33 that excluded. The authors have two points, but we don't see an explanation anywhere and why they were excluded all those papers.

4.     The sections 3.2 and 3.3 are written very good and I congratulate the Authors for that.

5.     The tables are clear and the report of the data. I don’t understand or unless I have misunderstood that why the Authors’ separated the L. Cavanna 2011 study and furthermore it is not clear why this was written, even the description in 3.1 in the last lines are making some references without soundness why they finally included this study since it not grouped according to the criteria set.

6.     It is imperative to address the significant degree of variability in the discussion and explain why the Authors using different methodologies? The sample is diverse? Describe the factors that affected this problem.

7.     Many studies include the value <1> in their confidence interval measurements which affects the reliability of the results. Furthermore, apart from the first measurement, the other two measurements in the meta-analysis have 0 in the random confidence interval and for that reason 60+, even if a p level is reported less than 0.01 the specific element should not be taken into account as statistically strong.

8.     The Authors have correctly analyzed their problematic in the discussion, it would be good if they could relate it a little more to the existing knowledge they have in the previous studies of the introduction.

Thank you

Minor editing of English language required

Author Response

Thank you very much for the expert review of our manuscript. We are now providing our point-to-point responses to the Reviewer 1's comments as follows.

Reviewer 1

Dear Authors,

I read your work entitled “Complications of Central Venous Access Devices Used in Palliative Care Setting for Terminally-ill Cancer Patients: A Systematic Review and Meta-analysis” and here I enclose my recommendations to you:

  1. The introduction is simple, short and clearly written and very nicely reports the existing knowledge around this topic.

Our reply: Thank you so much for your appreciation. We sincerely hope that our revised manuscript is also satisfactory and acceptable.

  1. The methodology is asserted since the Authors work was approved by PROSPERO. In order the Authors to enhance their work I suggest them to enter in their methodology the PICOS system as well.

Our reply: Thank you very much again for your nice suggestion. We have also included the methodology into the PICOS framework in the Materials and Methods section and the Supplementary Materials. We hope that the revised manuscript is satisfactory and acceptable.

  1. It would be good for the reader if the Authors would briefly describe the detection of the articles as they are in prism because they start the description from the 41 that reviewed and do not give details about the first part like the 33 that excluded. The authors have two points, but we don't see an explanation anywhere and why they were excluded all those papers.

Our reply: Thank you so much for your nice suggestion. We have provided an update about the reason of exclusion of articles in our systematic review and literature search in the revised manuscript Section 3.1 Search Strategy Results and also updated our Figure 1 for easy reference. I hope this will be more acceptable to the reviewer.

  1. The sections 3.2 and 3.3 are written very good and I congratulate the Authors for that.

Our reply: Thank you so much for your appreciation.

  1. The tables are clear and the report of the data. I don’t understand or unless I have misunderstood that why the Authors’ separated the L. Cavanna 2011 study and furthermore it is not clear why this was written, even the description in 3.1 in the last lines are making some references without soundness why they finally included this study since it not grouped according to the criteria set.

Our reply: We are terribly sorry for the mistakes made in the manuscript. The study conducted by Cavanna et al on central line insertion was included in our systematic review and meta-analysis (Reference 24 in our manuscript), while the other 4 studies (Reference 20 to 23 in our manuscript) focused on PICC insertion. We are sorry that we have not made it clear in Section 3.1 Search Strategy Results and now we have rectified it already. We sincerely hope that it is now more acceptable to the reviewer.

  1. It is imperative to address the significant degree of variability in the discussion and explain why the Authors using different methodologies? The sample is diverse? Describe the factors that affected this problem.

Our reply: Thank you very much for your nice comment. The variability arises from the small number of publications with limited geographical representation. After conducting the I2 and τ2 test, the random effect model is therefore chosen in view of high heterogeneity. We have provided more explanation in the methodology and the discussion part. We hope that the revised manuscript is satisfactory and acceptable

  1. Many studies include the value <1> in their confidence interval measurements which affects the reliability of the results. Furthermore, apart from the first measurement, the other two measurements in the meta-analysis have 0 in the random confidence interval and for that reason 60+, even if a p level is reported less than 0.01 the specific element should not be taken into account as statistically strong.

Our reply: Thank you for your comments.

  1. Our study presented the rates (per 100 observations) of catheter-related bloodstream infection (CRBSI) and thromboembolism (TE). The confidence intervals of the rates reported by the selected studies may include “1” if very few events observed.
  2. We have revised the tone of our conclusion. Please see the following edits in the revision.

In Section 3.3, “The overall complication rate including both CRBSI and TE yield a pooled estimate of 7.02% (95% CI 0.27–19.10) for PICC, which was higher than 1.44% (95% CI 0.30–4.14, p = 0.002) for central line (Figure 2). PICC had a higher complication rate of CRBSI (pooled estimate 2.03%, 95% CI 0.00–9.62) when compared to central line (pooled estimate 0.96%, 95% CI 0.12–3.41%, p = 0.046).”

In Discussion, we removed the statement “The findings were statistically significant and risk of thromboembolism being margin-ally significant.”

As a limitation, we mentioned that “the sample size of studies that investigated PICC were fewer than 40. Some studies did not observe CRBSI or TE and the corresponding (pooled) confidence intervals may include 0. This small sample size issue could be a concern that every observed event may affect the outcome substantially (e.g., the proportion is changed by 5% when the sample size was 20). The findings should be interpreted with cautions.”

  1. The Authors have correctly analyzed their problematic in the discussion, it would be good if they could relate it a little more to the existing knowledge they have in the previous studies of the introduction.

Our reply: Thank you very much for your valuable comments. We have made a bit more elaboration in 4. Discussion section in the revised manuscript to echo what we stated in the Introduction section and identified the challenges and difficulties of implementing CVAD insertion in the real-world setting. We sincerely hope that this will be more acceptable.

We have also performed English editing of our manuscript, and hope that it is now more comprehensible.

Thank you very much once again for your expert review.

Reviewer 2 Report

This is an interesting systematic review and meta analysis on complications of central venous access devices used in palliative care setting for terminally-ill cancer patients. The paper is well-written. I have several comment to improve the manuscript further:

1. "In contrast, PORT is the most technically demanding and expansive option leading to its low usage" There is a typo, it should be expensive instead of expansive?

2. While the databases used for the search are thoroughly listed, the authors should clarify whether the same search terms and criteria across all databases.

3. The title "Survival analysis" is potentially misleading. From the description provided, this section seems more about statistical analyses and comparisons related to complications rather than survival.

4. Ensure that the criteria for including complications in the statistical analysis is clear, and perhaps consider elaborating a bit more on how you decided what complications to include.

5. The mention of "Leave-one-out meta-analysis" is excellent, as it indicates robustness testing. But for the reader's benefit, the authors should consider to provide a brief explanation or reference about what it entails and why it's important.

6. The authors should be clearer on the inclusion and exclusion criteria of their review

7. The authors should report the inter-rater reliability of the abstract screening, full-text screening, and data extraction

8. I am surprised that there is no quality assessment conducted in this systematic review and meta-analysis. This is an important element in a systematic review and meta-analysis.

9. The acknowledgment of limitations is thorough. One potential addition might be to discuss the geographic spread of the studies the authors included – were they all from one region or globally dispersed? This might have implications for the generalizability of the findings.

10. Consider incorporating a brief mention of the practical or clinical implications

Author Response

Thank you very much for the expert review of our manuscript. We are now providing point-to-point responses to the reviewers’ comments as follows.

Reviewer 2

This is an interesting systematic review and meta-analysis on complications of central venous access devices used in palliative care setting for terminally-ill cancer patients. The paper is well-written. I have several comments to improve the manuscript further:

  1. "In contrast, PORT is the most technically demanding and expansive option leading to its low usage" There is a typo, it should be expensive instead of expansive?

Our reply: Thank you so much for your nice comment. I am very much sorry for the typo which was not detected in our spell check. We have now corrected it for your reference.

  1. While the databases used for the search are thoroughly listed, the authors should clarify whether the same search terms and criteria across all databases.

Our reply: Thank you very much for your nice update. We have confirmed and declared that the same search terms and criteria were adopted when we performed literature search across all databases.

  1. The title "Survival analysis" is potentially misleading. From the description provided, this section seems more about statistical analyses and comparisons related to complications rather than survival.

Our reply: Thank you so much for your valuable comment. Again, we are terribly sorry for the typo which should be “2.2 Statistical analysis”. We have rectified it in the revised manuscript accordingly.

  1. Ensure that the criteria for including complications in the statistical analysis is clear, and perhaps consider elaborating a bit more on how you decided what complications to include.

Our reply: Thank you very much for your nice suggestion and valuable comments. We have chosen overall complication rate, rate of catheter-related bloodstream infection (CRBSI) and rate of thromboembolism (TE) as the endpoints in our systematic review and meta-analysis. It is because these 3 complications are the most commonly reported CVAD-related complications in the literature and also the most clinically relevant complications associated with CVAD. We have further elaborated this in our revised manuscript in the 2.2 Statistical analysis and sincerely hope that it is now more acceptable to the reviewer.

  1. The mention of "Leave-one-out meta-analysis" is excellent, as it indicates robustness testing. But for the reader's benefit, the authors should consider to provide a brief explanation or reference about what it entails and why it's important.

Our reply: Thank you very much for your nice comment. The explanation has been added to the manuscript. We hope that the revised manuscript explains it clearer. Please see the below explanation in Discussion of the manuscript: “A publication is considered to be influential when its removal leads to the pooled estimate to fall outside the range of 95% confidence interval. The relevant study should be screened to look for an explanation to the discrepancies which included the study design and subject selection.”

  1. The authors should be clearer on the inclusion and exclusion criteria of their review.

Our reply: Thank you very much for your nice suggestion. We have further elaborated our inclusion and exclusion criteria of our systematic review in the revised manuscript and supplementary appendix, and added a PICOS framework to further explain our research questions as suggested by Reviewer 1. We sincerely hope that our further revision is satisfactory and acceptable.

  1. The authors should report the inter-rater reliability of the abstract screening, full-text screening, and data extraction

Our reply: Thank you very much for your nice suggestion. Of the 41 publications screened for eligibility, there was only 1 publication with doubtful eligibility which was then resolved by discussion and consensus. Thus the inter-rater reliability was . It was reported in Section 3.1 Search Strategy Results of the revised manuscript.

  1. I am surprised that there is no quality assessment conducted in this systematic review and meta-analysis. This is an important element in a systematic review and meta-analysis.

Our reply: Thanks for your valuable comment. We have added the quality assessment of the selected studies based on the Newcastle-Ottawa Scale (NOS). We included a section to describe NOS. Please see section 2.2 in the revised version.

“2.2 Quality assessment of selected studies

We referred to the Newcastle-Ottawa Scale (NOS) to assess the quality of the selected studies (https://www.ohri.ca/programs/clinical_epidemiology/oxford.asp). The NOS assess the quality of the studies based on three domains, namely, (a) the selection of the study groups, (b) the comparability of the groups, and (c) the ascertainment of the outcome of interest for cohort studies. In general, a study having a score of 7 or above, out of a maximum of 9, would be considered of a high quality with a low risk of bias. Studies with a score of 6 or below may represent a considerable risk of bias.

We now present the results of the respective assessment in Section 3.1 and Table 1.

“Based on the Newcastle-Ottawa Scale (NOS), the quality selected studies generally awarded scores ranging from 7 to 9, suggesting that the selected studies were in high quality with a low risk of bias.”

We sincerely hope that our elaboration and revision is more acceptable and satisfactory to the reviewer.

  1. The acknowledgment of limitations is thorough. One potential addition might be to discuss the geographic spread of the studies the authors included – were they all from one region or globally dispersed? This might have implications for the generalizability of the findings.

Our reply: Thank you so much for your valuable suggestion. We have offered our comments in the Discussion section as one of the study limitations and hope that it will be acceptable to the reviewer.

  1. Consider incorporating a brief mention of the practical or clinical implications

Our reply: Thank you so much for your valuable suggestion. We have added a brief discussion on the practicality and clinical implications of CVAD insertion for palliative care settings in the real-world setting in Section 4 Discussion. We sincerely hope that our further elaboration is also acceptable to the reviewer.

We have also performed English editing of our manuscript, and hope that it is now more comprehensible.

Thank you very much once again for your expert review.

Round 2

Reviewer 2 Report

The authors have addressed all my comments well. I appreciate their efforts.